# SRAM-Based CIM Architecture Design for Event Detection [note 1]

**DOI:** 10.3390/s22207854

**Published:** 2022-10-16

**Authors:** Muhammad Bintang Gemintang Sulaiman, Jin-Yu Lin, Jian-Bai Li, Cheng-Ming Shih, Kai-Cheung Juang, Chih-Cheng Lu

**Affiliations:** Industrial Technology Research Institute, 195, Section 4, Zhongxing Road, Zhudong Township, Hsinchu 310401, Taiwan

**Keywords:** artificial internet of things, computing in memory, convolutional neural network

## Abstract

Convolutional neural networks (CNNs) play a key role in deep learning applications. However, the high computational complexity and high-energy consumption of CNNs trammel their application in hardware accelerators. Computing-in-memory (CIM) is the technique of running calculations entirely in memory (in our design, we use SRAM). CIM architecture has demonstrated great potential to effectively compute large-scale matrix-vector multiplication. CIM-based architecture for event detection is designed to trigger the next stage of precision inference. To implement an SRAM-based CIM accelerator, a software and hardware co-design approach must consider the CIM macro’s hardware limitations to map the weight onto the AI edge devices. In this paper, we designed a hierarchical AI architecture to optimize the end-to-end system power in the AIoT application. In the experiment, the CIM-aware algorithm with 4-bit activation and 8-bit weight is examined on hand gesture and CIFAR-10 datasets, and determined to have 99.70% and 70.58% accuracy, respectively. A profiling tool to analyze the proposed design is also developed to measure how efficient our architecture design is. The proposed design system utilizes the operating frequency of 100 MHz, hand gesture and CIFAR-10 as the datasets, and nine CNNs and one FC layer as its network, resulting in a frame rate of 662 FPS, 37.6% processing unit utilization, and a power consumption of 0.853 mW.

## 1. Introduction

DEEP neural networks (DNNs) have highly flexible parametric properties, and these properties are being exploited to develop artificial intelligence (AI) applications in various domains ranging from cloud computing to edge computing. However, the high computational complexity and high-energy consumption of CNNs trammel their applications, particularly in terms of hardware. Regarding hardware, various CNN accelerators have been proposed to address computing needs, but most of them are still based on the Von Neumann architecture, which requires substantial amounts of energy to transfer massive amounts of data between memory and processing elements. Transferring a DNN to an edge device remains challenging because of the high storage, computing, and power requirements. To overcome this challenge, numerous high-throughput, low-power devices have been proposed in recent years to reduce the time complexity of matrix–vector multiplications. Computing-in-memory (CIM) reduces the massive data movement by performing computation on the memory to avoid the Von Neumann bottleneck issue. Nevertheless, CIM-based accelerators still need to overcome challenges.

To reduce the storage and computational costs, many different model compression algorithms have been proposed. In this particular model, a quantization algorithm is used, which is one of the most used compression algorithms. In the quantization algorithm, the input and weight bit width is limited to reduce the computational complexity by using different types of quantizers. These types include binary [1], ternary [2], uniform [3,4,5], and non-uniform quantizers [6,7,8].

Our SRAM-based CIM accelerator design is proposed to detect the event with ultra-low-power consumption. A hierarchical AI architecture shown in Figure 1 below, and is promising to save system power in AIoT applications. In the low-power sensor module, information captured from the peripheral sensor such as the imager is pre-processed to 32 × 32 image size and then sent into the CIM-based accelerator for event detection to trigger the precision inference in the next stage. Therefore, the end-to-end system power can be optimized and saved by at least a 30% reduction. The adopted SRAM CIM macro [9] in this paper can accommodate 8192 × 8 bit (64 Kb) weights, and contains 8 partitions. The 16 input data are shared in 8 partitions. These perform the inner product with the activation of the weight group at the same time, and then eight results are generated in the next cycle.

This article is organized as follows. Section 2 introduces the background of model quantization, the SRAM CIM macro, and the CIM-based accelerator. Section 3 describes the proposed SRAM-based CIM accelerator architecture design. Section 4 describes the equation-based profiling tool. Section 5 presents the experimental results, and Section 6 concludes this article.

## 2. Background

### 2.1. Model Quantization

Deep neural networks (DNNs) have achieved remarkable accuracies in various domains of tasks, including computer vision [10], speech recognition [11], and NLP [12]. However, the DNN model usually has many parameters that lead to large storage overhead and high computation complexity. These problems make it challenging to apply models on edge devices such as FPGA, and computing-in-memory (CIM). Recent research on model quantization has been proposed to reduce the bit precision of weights and activations. These quantization techniques transform weights and activations into low-bit data structures. Several quantization techniques [13,14] have significantly compressed the storage of the model. However, prior works have shown that quantization schemes would greatly affect accuracy.

### 2.2. SRAM CIM Macro

Von Neumann’s architecture is based on the stored-program computer concept, where instruction data and program data are stored in the same memory. This design is still used in most computers produced today. However, the von Neumann architecture is famous for its bottleneck due to the relative ability between processing elements and memories when a large amount of data movement is taking place. The CNN processes both training and inference and frequently requires a large amount of data to perform data and parameter modifications. CIM has been widely known as the solution to this problem through its ability to perform computational operation and store its data in the same place. Many SRAM CIM macros have been proposed and designed based on different applications. Figure 2 shows the concept of a CIM macro. Qing et al. [15] proposed a 4 + 2T SRAM macro for embedded searching and CIM applications. Zhang et al. [16] proposed a machine learning classifier that was implemented in a 6T SRAM array. Si et al. [17] proposed a dual-split-control 6T SRAM CIM that can support a fully connected layer. Biswas and Chandrakasan [18] proposed a 10T Conv-SRAM for binary weight neural networks. The aforementioned CIM macro works have shown the result of CIM advantages, particularly in functionality and energy efficiency. In this work, we proposed 4-bit input and 8-bit weight per computing unit as our hardware architecture design. We adopted the design from the 6T-SRAM chip [9], and the algorithm followed its specifications.

### 2.3. CIM-Based Accelerator

There have been many studies on CIM-based accelerators according to the different types of CIM macros. The ReRAM-based accelerator is the most well known as it has less area and high energy efficiency while supporting on-chip training [19,20,21,22,23]. RECOM [24] was the first CIM-based accelerator to support DNN processing. Jiang et al. [25,26] proposed SRAM CIM accelerators for CNN training. However, in several cases, it was assumed that the CIM macros were in the ideal state. Few studies have designed a CIM macro for system-level architecture. The proposed CIM-based architecture in this work is aimed at the optimization of CNN inference to detect the event with ultra-low-power consumption. Moreover, this architecture is based on the taped-out CIM macro instead of the ideal one in the aforementioned cases. Several aspects differentiate the condition between the ideal and the taped-out CIM macro. The ideal CIM macro usually has a large capacity that does not bring multiple reloading during calculation, whereas the taped-out CIM macro has a limited capacity. The ideal CIM macro can complete high-precision calculations, as opposed to the taped-out macro with its low-precision calculations. The taped-out CIM macro must concern with the analog-to-digital converter (ADC) number and its variation caused by the BL current [27]. Therefore, this work is proposing a hardware and software co-design to compensate for the limitation of the CIM macro.

## 3. Proposed SRAM-Based Accelerator Architecture Design

### 3.1. CIM-Aware Quantization Algorithm

A linear quantization function is added to the model while training in our quantization scheme. The weight quantization function is as follows:(1)$$w_{fp}^{\prime }\leftarrow T_{w}\left(w_{fp}\right)$$
(2)$$w_{fp}^{q}\leftarrow 2\cdot \frac{round\left(w_{fp}^{\prime }\cdot \left(2^{b_{w}}-1\right)\right)}{2^{b_{w}}}-1$$
(3)f(x)=max(0,x)
where $w_{fp}$ and $w_{fp}^{q}$ represent the weight before and after quantization in a floating point scope. $T_{w}$ is a non-linear transformation function to restrict the range of weight. Equation (2) quantizes the transformed weights $w_{fp}^{\prime }$ to $2^{b_{w}}$ groups of data. The ReLU formula of Equation (3) is used as the activation function to restrict the range of activations. The quantized weights will be mapped to integers by a mapping function at the inference step. These make the matrix multiplication an integer-only arithmetic. We use hand gesture [28] and CIFAR-10 [29] datasets as our training and validation data. Both datasets have 32 × 32 image sizes, as shown in Figure 3. The ten classes of classification accuracy reach 99.70% and 70.58% with the proposed tiny model, respectively [30].

### 3.2. Top-Level Design

Figure 4 shows the proposed architecture including ping-pong SRAM and CIM macro. The chip architecture consists of two main blocks, which are the memory unit and the CIM unit. The memory unit consists of two identical 64 kb SRAMs (SRAM A and SRAM B). The controller handles the instruction code and controls the operation of the chip. At first, SRAM A acts as an input SRAM that receives the input data off-chip. The scheduler then arranges the data movement from SRAM into the CIM unit. The arrangement block handles the movement of the convolution output from the CIM unit into SRAM. The CIM macro receives the weight data and handles the convolution process (multiplication between input data and weight). In a normal convolution process, the convolution output shall be transmitted from SRAM into the off-chip, and then the input of the next layer will be transmitted from the off-chip again. This has the drawback of the latency of sending data on-chip and off-chip through the interface. The interface is the gate between on-chip and off-chip data movement, mainly handling the data flow from off-chip such as input data, weight, and transfer convolution output to off-chip. Therefore, our memory system unit adapts the ping-pong SRAM mechanism to decrease the latency. Instead of sending the convolution output off-chip, the receiving SRAM at the current layer will act as the input SRAM in the next layer. Finally, the last layer of convolutional output will be sent off-chip to perform the computation of the fully connected layer.

The CIM core mainly contains two SRAM CIM macros (also acting as a weight SRAM), an accumulator, and an activation block. Because of CIM’s parallel computing with favorable efficiency, conventional digital PE is replaced with CIM macros. Taking advantage of SRAM CIM being both memory and PE, the energy consumption of the transfer weight between the memory and the PE is minimized. Weight SRAM accommodates the weight of the current layer because CIM cannot store all weights at once; therefore, the CIM macro must reload new weights for each new layer. The result of the internal calculation of the CIM is accumulated by the shift accumulator and the kernel accumulator. The accumulated result is further processed by the activation block, and the calculated result is stored in the OFM SRAM.

### 3.3. Weight Data Mapping

The design of the adopted SRAM CIM macro accommodates the maximum capacity of 8192 × 8-bit (64 kb) weights and contains 8 partitions. Each of these partitions are divided into 64 groups of 16 weights. These 16 weights of a group are defined as a weight group. The details of these partitions are described in Figure 5a. When using a SRAM CIM macro for computing, each partition activates one weight group at the same relative position through the control signal. All weights of a kernel must be stored in SRAM CIM macros. Therefore, mapping weight to the SRAM CIM mostly follows the sequence according to kernel size, as shown in Figure 5b. The SRAM CIM macro contains 8 partitions, and each partition contains 64 weight groups. Each grid represents 1 weight group, and the 16 weight groups in each partition at the same position are considered GS. A GS is defined as a weight group at the same position on each kernel of each subfilter. If α = 16 and N = 16, GS1-1 in Figure 5b denotes the GS which contains K1-1, K2-1,…, K16-1, with a total of 256 weights.

The 16 input data contain 4-bit data each, are passed into a demultiplexer [31], and shared in 8 partitions. These perform the inner product with the activation of the weight group at the same time, and then eight results are generated in the next cycle. This inner product operation behavior is in accordance with the convolution calculation in the CNN, as shown in Figure 6. In this work, the two SRAM CIM macros sharing the same control signal and input are combined into one core to acquire higher parallel computation capability. By doing so, the 16 weight groups can be activated at the same time and perform 16-vector inner products of 16 kernels in one cycle. These 16 weight groups at the same relative position are defined as a group set (GS). The parallel computing feature of CIM can be viewed as eight convolution operations at one time. Figure 6 is an example of an IFM convolving with three 3 × 3 × 16 kernels and mapping these kernels onto different CIM partitions. K1-1–K1-9 denote the weight groups of kernel 1, while each weight group contains 16 weights in the channel direction.

## 4. Equation-Based Profiler

### 4.1. Configuration Parameter

After we proposed the design of the SRAM-based CIM architecture, we proceed to perform the analysis of the estimation result of the design. A profiling tool is important for performing an analysis of the source and target data structures for data integration. The configuration parameters on the profiler are based on the parameter from the architecture design. The configuration parameters are operating frequency (*OF*), input and output bandwidth (*IOB*), CIM input (*CI*), convolution output (*CO*), bit representation (*BR*), maximum weight capacity (*MWC*), and SRAM size. The default operating frequency is 100 MHz, while further experimental results will include the range between 10 and 100 MHz with the step of 10 MHz. The input and output bandwidth is set at 16 bits. The CIM input and convolution output represent the amount of the input and output channel from the CIM macro, and those are set at sixteen and eight channels, respectively. The bit representation, meaning the quantized n-bit being used for the inference process, is set at 4 bits. The maximum weight capacity is the maximum value of weight data distribution on the CIM macro bank, and is set at 32 kb. Lastly, the SRAM size being used is 64 kb with a combined size of 1024 words and 64 bits (one word is 4 bits in size).

### 4.2. Network Parameter

The parameters for the profiling tool are provided based on the network being used. In our case, we use nine CNN layers and one fully connected layer with image sizes in 32 × 32 pixels. The parameters consist of input shape, zero padding, stride, and output shape, which are presented in Table 1. As for the input image, the parameter for the input shape at layer 1 will be 32 pixels height times 32 pixels width, times three channels. We use notations such as *H*, *W*, and *I* to denote the height, width, and channel for input shape, respectively. Filter or kernel defines the size of the convolution matrix denoted by its height (*R*) and width (*S*). Zero padding (*Z*) refers to the usage of the zero-padding feature on each layer. It returns TRUE if the corresponding layer uses zero padding or FALSE otherwise. Stride uses the step of 1 or 2 with the configuration of vertical (*V*) and horizontal (*L*) stride as 1 × 1 or 2 × 2. The output shape defines the size of the output result after the convolution process. The filter size and stride combination will produce a different output shape result. The output shape consists of its height (*M*), width (*N*), and channel (*O*).

### 4.3. Data Size Calculation

The data size calculations represent how much data is being used during the dataflow and convolution process, as shown in Table 2. Dataflow consists of the data movement of both off-chip and on-chip. The input data size indicates the data size for the input feature map, as shown in Equation (4). These data are being moved through the interface from the off-chip into the SRAM. The equation for input data size involved the input shape (*I*, *H*, *W*) on the corresponding layer with the bit representation (*BR*). On layer 1, if the zero padding is TRUE, then the input channel (*I*) uses 16 channels. The output data size indicates the size for the output feature map (after the convolution process), as shown in Equation (5), with the data being moved off-chip. This equation involves the output shape (*M*, *N*, *O*) on the corresponding layer with the bit representation (*BR*). On the FC layer, bit representation is 1 bit. The weight data have the movement from off-chip into the CIM macro, with the size calculated as in Equation (6). It can be seen that this equation consists of input shape (*I*, *H*, *W*), output shape channel (*O*), and bit representation (*BR*). The calculation in Equation (7) below is how many processes happened during the MAC (multiply and accumulation) process on convolution.
(4)Input Data Size=I×H×W×BR
(5)Output Data Size=M×N×O×BR
(6)Weight Data Size=I×H×W×O×BR
(7)Calculation=I×R×S×M×N×O×2

### 4.4. Data Cycles and MAC Cycles

The input, weight, and output data cycles are the amounts of the cycles required during the IFM, weight, and OFM data movement, respectively. While the MAC cycle is the amount of the cycle during the calculation process, the input data cycle involves the input data size (*IDS*), I/O bandwidth (*IOB*), bit representation (*BR*), and 12 as the constant. The output data cycle has a similar configuration, except it uses output data size (ODS) instead of input. The weight data cycle equation required maximum weight capacity (*MWC*), weight data size (*WDS*), and bit representation (*BR*). The MAC cycles equation involves input channel (I), stride (V, L), output channel (O), filter/kernel (R, S), zero padding (Z), CIM input (*CI*), and convolution output (*CO*). TOTAL values represent the total cycles required for each layer; they are the summation of input, output, weight data, and MAC cycles. The cycle calculations are shown in Equations (8)–(11) and the results are shown in Table 3.
(8)$$Input\;Data\;Cycle=round\left(\frac{IDS}{IOB\;\times \;BR}\right)\times 12\;$$
(9)$$Weight\;Data\;Cycle=\frac{MWC-WDS}{BR}$$
(10)$$Output\;Data\;Cycle=round\left(\frac{ODS}{IOB\;\times \;BR}\right)\times 12$$
(11)$$MAC\;Cycles=\frac{\frac{\frac{I\;\times \;V\;\times \;L\;\times \;\left(Z\{\begin{array}{l} True,\;\frac{W}{L}\\ False,\;\frac{W\;-\;S\;+\;1}{L} \end{array}\right)\;\times \;\left(Z\{\begin{array}{l} True,\;\frac{H}{V}\\ False,\;\frac{H\;-\;R\;+\;1}{V} \end{array}\right)\;\times \;O\;\times \;R\;\times \;S}{L}}{V}}{CI\times CO}$$

## 5. Experimental Results

This section shows the results of the profiling from the proposed SRAM-based CIM architecture design. The experimental results consist of quantization result, frame rate, MAC utilization, power consumption, and energy consumption per inference.

### 5.1. CIM-Aware Quantization Algorithm Result

To obtain the most efficient accuracy result among the quantized bit width, we compare the model using two different datasets: hand gesture and CIFAR-10. Hand gesture datasets include 10 classes of 32 × 32 grayscale images, while CIFAR-10 datasets consist of 10 classes of 32 × 32 RGB images. All the datasets are being trained using the VGG9 model (9CNN + 1 FC layer), with the first layer and FC layer computation performed in FP32. The accuracy result for bit width with the weight and activation bits of 8b/32b, 8b/8b, and 8b/4b, respectively, are shown in Table 4 below. Among the accuracy performance of the quantized bit activation, the 4-bit activation receives an acceptable/reasonable error drop compared to the full precision (32 bit) and 8 bit.

### 5.2. Frame Rate

Frame rate is the frequency or rate at which consecutive images (also called frames) are captured or displayed. It is usually also expressed as frames per second or FPS. To calculate the frame rate, we need to obtain the total cycles and the operating frequency. The total cycles are obtained from the previous calculation in Section 4.4, and it is the summation of the data and MAC cycles from layer one (Conv1) to the last layer (FC). Once we obtain the total cycle, we can obtain the total time by dividing it by the operating frequency. As for the frame rate, we can calculate it by dividing 1 by the total time. The chart in Figure 7 shows the frame rate for various operating frequencies from 10 to 100 MHz with three different image sizes.
(12)$$Total\;Time=\frac{Total\;Cycles}{Operating\;Frequency}\;$$
(13)$$Frame\;Rate=\frac{1}{Total\;Time}\;$$

### 5.3. MAC Utilization

MAC utilization indicates how much the operation cycles occupy the whole inference process, including data movement. This can be obtained by dividing the MAC cycles by the total cycles. Figure 8 shows the MAC utilization for different image sizes.
(14)$$MAC\;Utilization=\frac{MAC\;Cycles}{Total\;Cycles}\;$$

### 5.4. Power Consumption

Power consumption measurements can be performed for an exploratory purpose to understand and study the power consumption profiles of the proposed design. Power is measured in Watts, a unit of power in the International System of Units (SI) equal to one joule of work performed per second. Operation per cycle is defined by dividing the calculation by MAC cycles. It can be calculated in a layer or the total for all layers and will return the same result. *P_total_* is the power consumption measured at the given power rating. The power rating is based on the assumption of an ideal power rating, and it is obtained at 30 TOPS/W at a 100 MHz operating frequency. *P_total_* calculation can be seen in Equation (16), with the terra unit value being 10^12^ (as in 1 Terra = 10^12^). Power consumption in a mixed-signal CMOS circuit can be briefly divided into two components baswed on the Equation (17). *P*_static_ is composed of leakage, bias current in analog circuit, etc. (fixed across different frequencies). *P_dynamic_* occurs due to the transient current when switching the CMOS digital circuit (proportional to frequency). The power consumption for different operating frequencies can be measured by using Equation (18), and the result is shown in Figure 9.
(15)$$Operation\;per\;Cycle=\frac{Calculation}{MAC\;Cycles}$$
(16)$$P_{total}=\frac{\frac{\left(Operating\;Frequency\right)\;\times \;\left(Operation\;per\;Cycle\right)}{Terra\;Unit}}{Power\;Rating}$$
(17)$$P_{total}=P_{static}+P_{dynamic}$$
(18)$$P_{total}@Freq=\left(P_{total}@100\;\mathrm{MHZ}\;\times \;P_{Static}\right)+\left(P_{total}@100\;\mathrm{MHZ}\;\times \;P_{dynamic}\;\times \;\frac{Freq}{100\;\mathrm{MHZ}}\right)$$

### 5.5. Energy Consumption per Inference

Energy consumption per inference is measured to calculate how much energy is required in the proposed design system to perform one cycle of the inference process. The energy consumption per inference can be obtained by using Equation (19) below. We compare the result of the energy consumption of inference across different operating frequencies and image sizes. Figure 10 shows the comparison of its utilization.
(19)$$Energy\;Consumption\;per\;Inference=\frac{Power\;Consumption\times Total\;Cycles}{Operating\;Frequency}$$

## 6. Discussion

We use the CIM (computing-in-memory) mechanism to overcome the Von Neumann bottleneck issue. The bottleneck is mainly summed up by three aspects: data movement between memory arrays and the processing unit results in non-negligible latency; data movement in memory hierarchies is greatly limited by bandwidth; high energy consumption, such as the power consumption of moving data between computing and off-chip memory units, is 100 times more than floating point computing. To overcome such problems, CIM technology is proposed. The key idea of the proposed technology is to bring memory and computing closer instead of separating them, therefore improving the efficiency of the data movement. Our proposed model is based on the difference between the ideal CIM macro and the taped-out one. The ideal CIM macro usually has a large capacity that does not bring multiple reloading during calculation and is able to perform high-precision calculations. However, the taped-out CIM macro has a limited capacity, has low-precision calculations, and must regard the analog-to-digital converter (ADC) number and its variation caused by the BL current. The proposed CIM-based architecture in this work is aimed at optimization based on the limitation of the taped-out CIM macro by proposing a hardware and software co-design.

Our profile tool is very simple and restricted (narrowed to our architecture). One of the well-developed profilers for the CIM hardware accelerator is NeuroSim [32]. However, the reason why we do not use the existing profiling tool is that our architecture design focus is on CIM macro development. Moreover, several inputs to the simulator are different, including memory types, nonideal device parameters, transistor technology nodes, network topology and sub-array size, and training datasets and traces. None of the other CIM simulators have been validated with the actual silicon data (although NeuroSim has been validated with SPICE simulations using the PTM model and FreePDK. It is known that the PTM model and FreePDK are for educational purposes rather than for foundry fabrication purposes).

We can explore further the algorithm, data movement, and circuit design perspective to reduce the computational cost in the future. From the algorithm perspective, in the current state, we only applied the quantization method. In the future, we are planning to use the pruning algorithm to enable the sparsity of connections. Therefore, we can reduce the data movement due to the lesser connections. This also makes it possible to achieve a reduction in area and energy consumption in the circuit design [33].

## 7. Conclusions

This article proposed a software and hardware co-design to design a CIM-aware model quantization algorithm and an SRAM-based CIM accelerator. In the design, the CIM-aware algorithm with 4-bit activation and 8-bit weight is examined on hand gesture and CIFAR-10 datasets, and determined to have 99.70% and 70.58% accuracy, respectively. A profiling tool to analyze the proposed design is also developed to measure how efficient our architecture design is. The proposed design system utilizes the operating frequency of 100 MHz, hand gesture and CIFAR-10 as the datasets, and nine CNNs and one FC layer as its network, resulting in a frame rate of 662 FPS, 37.6% processing unit utilization, and power consumption of 0.853 mW.

## Figures and Tables

**Figure 1 sensors-22-07854-f001:**
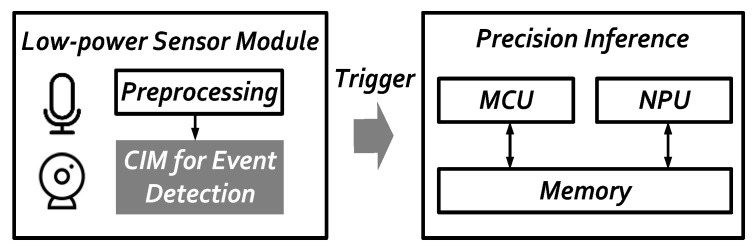
Hierarchical AI architecture.

**Figure 2 sensors-22-07854-f002:**
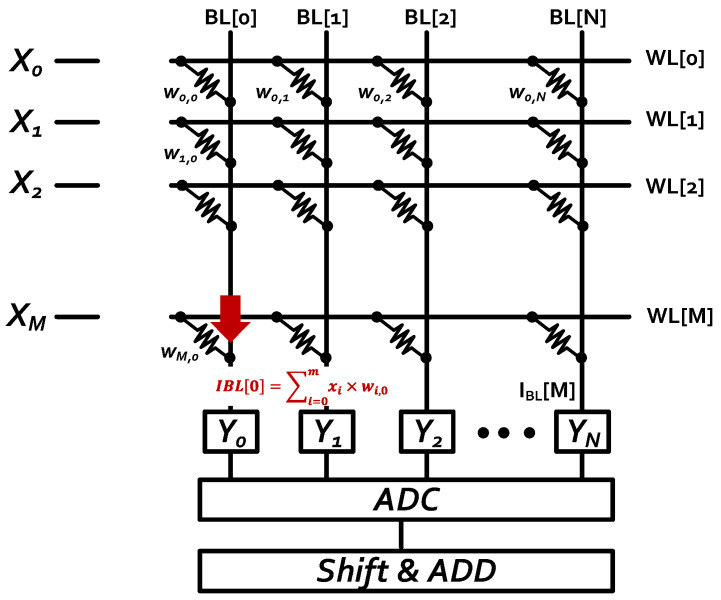
The concept of CIM macro.

**Figure 3 sensors-22-07854-f003:**
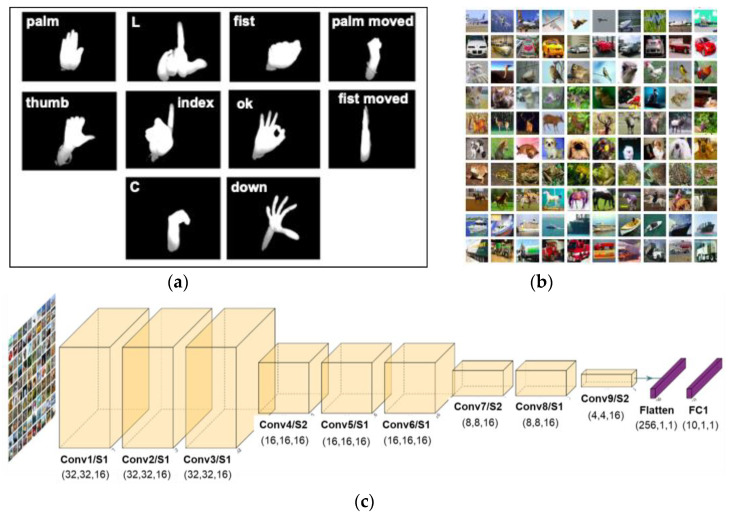
(**a**) Hand gesture datasets; (**b**) CIFAR-10 datasets. (**c**) The developed tiny CNN model.

**Figure 4 sensors-22-07854-f004:**
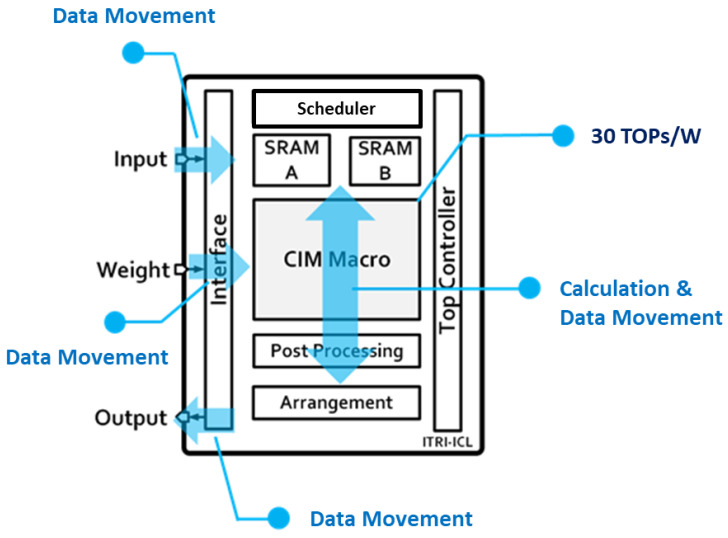
SRAM-based CIM chip architecture design.

**Figure 5 sensors-22-07854-f005:**
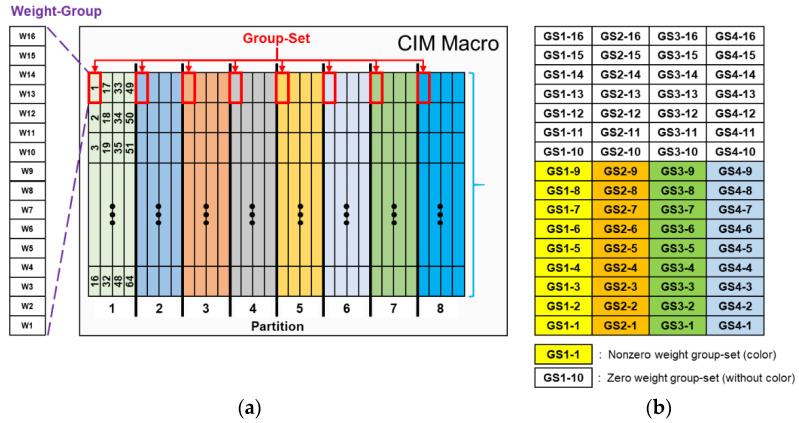
(**a**) SRAM CIM macro partitions. (**b**) Weight mapping in one partition of the CIM, showing the mapping situation that contains both zero and non-zero weight groups of the CIM.

**Figure 6 sensors-22-07854-f006:**
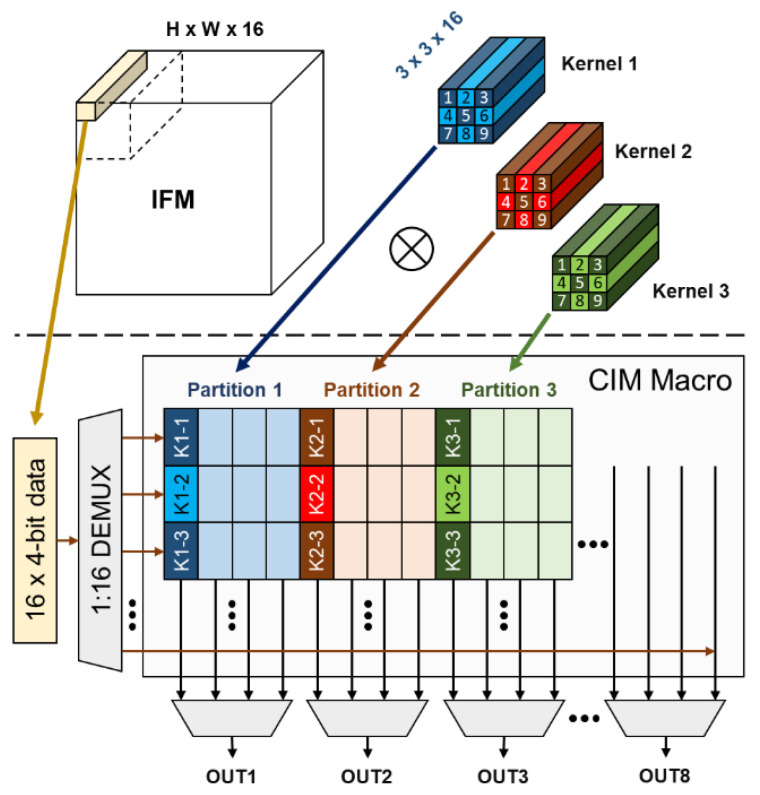
Example of an IFM convolving with three 3 × 3 × 16 kernels and mapping these kernels onto different CIM partitions.

**Figure 7 sensors-22-07854-f007:**
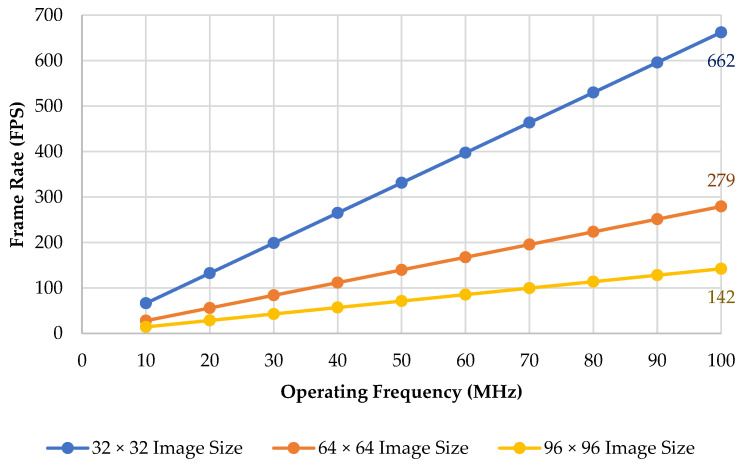
The frame rate across the different operating frequencies.

**Figure 8 sensors-22-07854-f008:**
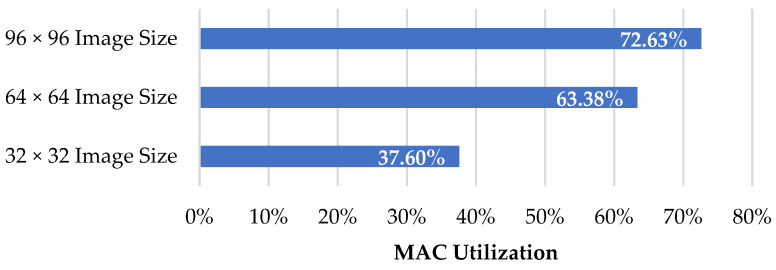
MAC utilization for different image sizes.

**Figure 9 sensors-22-07854-f009:**
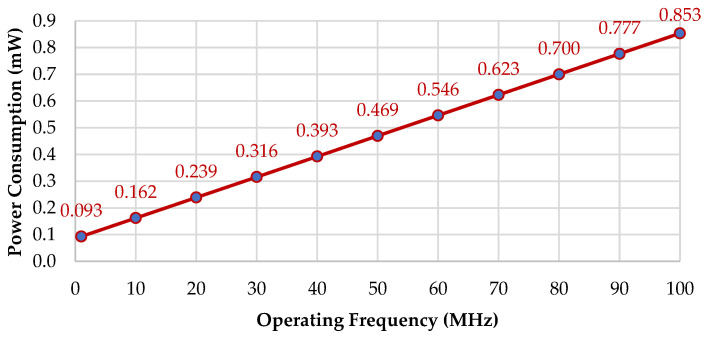
Power consumption across the different operating frequencies.

**Figure 10 sensors-22-07854-f010:**
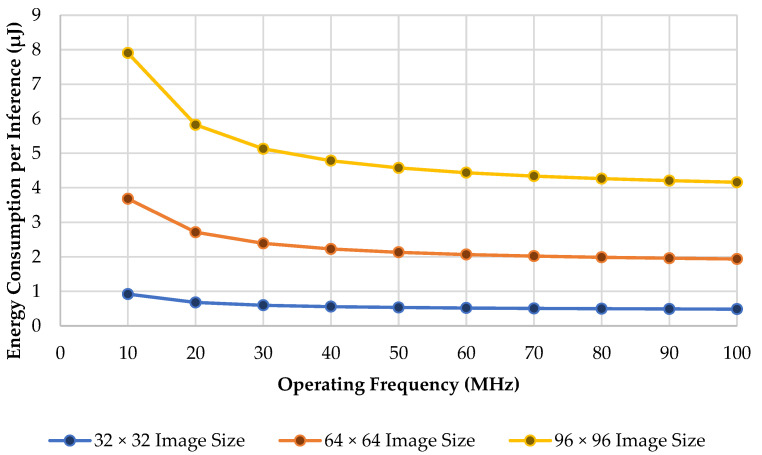
Utilization of each convolution layer.

**Table 1 sensors-22-07854-t001:** Network parameters using the configuration of 9 CNNs and 1 FC layer.

Layer	Input Shape			Filter/Kernel		Zero Padding	Stride		Output Shape		
	Height	Width	Channel	Height	Width		Vertical	Horizontal	Height	Width	Channel
	H	W	I	R	S	Z	V	L	M	N	O
Conv1	32	32	3	3	3	TRUE	1	1	32	32	16
Conv2	32	32	16	3	3	TRUE	1	1	32	32	16
Conv3	32	32	16	3	3	TRUE	1	1	32	32	16
Conv4	32	32	16	3	3	TRUE	2	2	16	16	16
Conv5	16	16	16	3	3	TRUE	1	1	16	16	16
Conv6	16	16	16	3	3	TRUE	1	1	16	16	16
Conv7	16	16	16	3	3	TRUE	2	2	8	8	16
Conv8	8	8	16	3	3	TRUE	1	1	8	8	16
Conv9	8	8	16	3	3	TRUE	2	2	4	4	16
FC	4	4	16	1	1	FALSE	1	1	4	4	10

**Table 2 sensors-22-07854-t002:** Data size calculation based on the configuration and network parameters.

Layer	Input Data Size	Weight Data Size	Output Data Size	Calculation
	IDS	WDS	ODS	MAC
Conv1	65,536	1728	65,536	884,736
Conv2	65,536	9216	65,536	4,718,592
Conv3	65,536	9216	65,536	4,718,592
Conv4	65,536	9216	16,384	1,179,648
Conv5	16,384	9216	16,384	1,179,648
Conv6	16,384	9216	16,384	1,179,648
Conv7	16,384	9216	4096	294,912
Conv8	4096	9216	4096	294,912
Conv9	4096	9216	1024	73,728
FC	1024	640	160	5120

**Table 3 sensors-22-07854-t003:** Data and MAC cycles based on the configuration and network parameters.

Layer	Input Data Cycles	Weight Data Cycles	Output Data Cycles	MAC Cycles	TOTAL
Conv1	12,288	8192	0	3456	23,936
Conv2	0	8192	0	18,432	26,624
Conv3	0	8192	0	18,432	26,624
Conv4	0	8192	0	4608	12,800
Conv5	0	8192	0	4608	12,800
Conv6	0	8192	0	4608	12,800
Conv7	0	8192	0	1152	9344
Conv8	0	8192	0	1152	9344
Conv9	0	8192	0	288	8480
FC	0	8192	30	42	8264

**Table 4 sensors-22-07854-t004:** Quantization comparison results between datasets.

Model	Bit Width	Accuracy (%)	
	W/A	CIFAR-10	Hand Gesture
VGG9	8b/32b	74.65	98.94
FP32 on 1st Conv & FC-layer	8b/8b	70.78	99.7
	8b/4b	70.58	99.7

## Data Availability

Publicity available datasets were analyzed in this study. Theses datas can be found here: https://www.kaggle.com/datasets/gti-upm/leapgestrecog (accessed on 2 May 2021) and https://www.cs.toronto.edu/~kriz/cifar.html (accessed on 1 April 2021).

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
