# Peer review of "SRAM-Based CIM Architecture Design for Event Detection†"

_sensors, 2022, doi:10.3390/s22207854_

Round 1

Reviewer 1 Report

Major:

Conclusion Section: The final results were stated clearly. However, there was not any comparison with previous literature to evaluate the significance of the proposed methodology.

Minor:

Figure 5:

     1. Hand Gestures figure is not clear enough; needs to be replaced with clearer examples from the used data set. 

     2. The CNN model configuration’s text is small and not clear; needs to be much clearer and easier to read.

Reviewer 2 Report

This paper designed a hierarchical AI architecture to optimize the end-to-end system power in the AIoT application. Comments to authors are list below:

1. The proposed SRAM CIM architecture design is too simple and related techniques only have a brief description; no novelty is shown in this article. For example, what’s the novelty and contribution of the quantization algorithm; the ping-pong SRAM buffer is also a well-known scheme, only this technique is not sufficient further reduce on-chip and off-chip data movement in comparison with related architecture designs. All these works of this paper seem only an implementation of existing techniques.

2.   In addition, the algorithm for weight data mapping is too rough and I wonder that authors have study the existing dataflow and mapping techniques for CIM.

3.   The proposed CIM architecture seems can only apply for the network with 16 channels in all CNN layers, which is not make sense.

4.    The profile tool is also very simple and restrict to narrow and fixed architecture, there have existing many well develop profile and simulation tools.

5.      Discussion on the experiment results is poor.

6.   In summary, the works in this article is only an implementation of existing simple techniques on a simple architecture design, with application on fixed and restrict network architecture.

Reviewer 3 Report

1. Why does the author have preferred 4 bt activation? 2. How will the author reduce the computational cost in the future? 3. How will you find the effectiveness of the proposed model? 4. How does the proposed scheme overcome the Von Neumann bottleneck issue? 5. In table 3, total value represents what and mention how it is calculated?  6. What is the initial energy level and residual energy level? 7. Refer the latest and suitable references as follows: Bansal, Malti, Harmandeep Singh, and Gaurav Sharma. "A taxonomical review of multiplexer designs for electronic circuits & devices." Journal of Electronics 3, no. 02 (2021): 77-88. Madhura, S. "A Review on Low Power VLSI Design Models in Various Circuits." Journal of Electronics 4, no. 2 (2022): 74-81.

Round 2

Reviewer 2 Report

Although the proposed methodologies still restrict to the limitation of taped-out CIM macro, this revision has made improvement and addressed my concerns satisfactorily.

Reviewer 3 Report

Revised version is accepted